# Bridge Health Monitoring Using Proper Orthogonal Decomposition and Transfer Learning

**Samira Ardani** [1,*] **, Saeed Eftekhar Azam** [2] **and Daniel G. Linzell** [1]

1. Department of Civil and Environmental Engineering, University of Nebraska-Lincoln, Lincoln, NE 68588, USA
2. Department of Civil and Environmental Engineering, University of New Hampshire, Durham, NH 03824, USA
* Correspondence: samira.ardani@unl.edu

**Featured Application: Transfer Learning (TL) in structural health monitoring is used for generalizing the trained knowledge for damage identification of a group of similar structures. TL significantly reduces the computational cost associated with retraining Machine Learning (ML) algorithms.**

**Abstract:** This study focuses on developing and examining the effectiveness of Transfer Learning (TL) for structural health monitoring (SHM) systems that transfer knowledge about damage states from one structure (i.e., the source domain) to another structure (i.e., the target domain). Transfer Learning (TL) is an efficient method for knowledge transfer and mapping from source to target domains. In addition, Proper Orthogonal Modes (POMs), which help classify behavior and health, provide a promising tool for damage identification in structural systems. Previous investigations show that damage intensity and location are highly correlated with POM variations for structures under unknown loads. To train damage identification algorithms based on POMs and ML, one generally needs to use multiple simulations to generate damage scenarios. The developed process is applied to a simply supported truss span in a multi-span railway bridge. TL is first used to obtain relationships between POMs for two modeled bridges: one being a source model (i.e., labeled) and the other being the target modeled bridge (i.e., unlabeled). This technique is then implemented to develop POMs for a damaged, unknown target using TL that links source and target POMs. It is shown that the trained knowledge from one bridge was effectively generalized to other, somewhat similar, bridges in the population.

**Keywords:** structural health monitoring; Transfer Learning; domain adaptation; damage detection; Proper Orthogonal Decomposition; classification; bridge

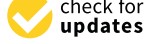



## 1. Introduction

Aging bridge populations compel owners to investigate effective, low-cost approaches for locating deficiencies across their inventory. Integrating automated structural health monitoring (SHM) and affiliated damage identification methods into bridge condition assessment and management processes could help bridge owners better manage their assets and maintain safe operations more cost-effectively. When dealing with SHM, Machine Learning (ML) methods are often used to predict damage location and intensity from measured or simulated structural response [1]. Proper Orthogonal Decomposition (POD) has proven to be a powerful method for extracting damage information from measured response [2]. However, this method, like most ML methods, is applicable to a single structure under specific damage scenarios [3]. In addition, obtaining ground truth information from real world structural damage states can be challenging, if not impossible. To alleviate these limitations, this study aims to use Transfer Learning (TL) as a tool for transferring knowledge about a feature space from a structure with pre-trained classifiers of damage location and intensity with known labels to another structure with unknown labels. The proposed method utilizes POD to detect damage and TL to generalize techniques to a

wide variety of bridges with some degree of similarity in behavior and damage states by transferring, a priori, knowledge from one structure to similar structures in the population.

Data-driven methodologies for damage identification in structures are often developed using ML and, more specifically, pattern recognition algorithms [4]. The most common approach for implementing data-driven damage identification is to find a robust damage feature from measured data. One of the main challenges in feature selection is finding features that are sensitive to damage and insensitive to other operational variables. Additionally, the correlation of feature behavior with damage level and potential low dimensionality of the feature vector are crucial factors affecting successful damage identification [5]. Although modal parameters are common features for most vibration-based damage identification methods, they cannot be directly used for damage identification in highly nonlinear structural systems [6]. In addition, modal parameters need to be decoupled from sensor noise to achieve improved damage detection accuracy [7]. Malekjafarian et al. [8] investigated the fault detection of an in-service railway track using measured acceleration. They demonstrated that, for a certain range of train forward speed, the extracted amplitude of acceleration after applying Peak-Based Decomposition (PBD) corresponded to the data observed from the Track Recording Vehicle (TRV) and could potentially be considered a damage indicator. Song et al. [9] proposed ensemble empirical mode decomposition (EEMD) to eliminate unnecessary information from the original signal for analyzing the pantograph–catenary system.

Principal Component Analysis (PCA), a POD method, is a powerful technique for capturing dominant features in a multi-dimensional system using a few modes [10]. To eliminate environmental effects, Bellino et al. [11] proposed a PCA-based damage identification approach that chose the first natural frequency as the damage feature and assessed its performance by examining an experimental time varying system under controlled temperature. Galvanetto and Violaris [12] numerically studied structural damage detection using Singular Value Decomposition (SVD) by computing Proper Orthogonal Modes (POMs) and examined differences between POMs in healthy and damaged structure models. Shane and Jha [13] developed a POD-based damage identification algorithm for a composite beam using vibration data. Eftekhar Azam et al. [14] developed an automated damage detection framework utilizing POD and Artificial Neural Networks (ANNs) to detect the location of simulated fatigue cracks in a steel railway bridge. More recently, Ardani et al. [15] examined the effectiveness of a POD framework for damage identification in three simply supported bridges by imposing actual damage scenarios.

The damage identification framework for complex structural systems is developed by training the damage features to be mapped to the corresponding damage labels. ANN is utilized to model this relationship between input (damage features) and output (damage labels) data. This algorithm has been widely used for pattern recognition and classification, and, in structural engineering, for the identification of deficiencies. Xu and Humar [16] proposed an ANN-focused, two-step algorithm that implemented modal energies for simulated damage identification using an FE girder bridge model. Mehrjoo et al. [17] demonstrated the efficacy of ANNs in identifying damage location and severity in truss bridge connections using modal shape parameters. Gu et al. [18] proposed using a multi-layer ANN that focused on changes in natural frequencies. Novelty indices that quantified damage severity were determined to distinguish changes in natural frequencies caused by damage from those caused by temperature variations.

Despite the large amount of research that successfully utilized ANNs as a learning method for applications of structural damage identification and SHM to various engineering problems, training time and output accuracy depend heavily on network structure. In many cases, the amount of computational effort required to retrain a developed network for a structure in a population with some degree of similarity to a structure within a trained network is excessive.

Finite Element (FE) modeling uncertainties (MUs) can lead to unreliable ANN results [19]. MU can be influential with respect to damage detection algorithm accuracy even if there is a good match between FE model predictions and experimental data [20]. Lee et al. [21] used differences between mode shape ratios for damage identification on a simple beam and a multi-girder bridge using ML to examine the effect of MU on damage detection. The proposed method was applied to an in-service, multi-girder bridge, and minor estimated damage intensity false positives were observed. Bakhary et al. [19] utilized statistical ANNs involving modal parameters from an FE model of a single span steel portal frame to investigate the effects of MU on vibration-based damage detection by considering random errors. They concluded that the statistical ANN detected damages with higher accuracy compared to a normal ANN. Rageh et al. [22] investigated the effects of MU on an operational railway bridge using a hybrid damage detection algorithm based on POD and ANNs. A series of numerical investigations were completed that involved different MUs and a robust damage feature was developed that was less sensitive to MUs. Result accuracy varied based on examined MU, with less accurate results being obtained at higher MU levels.

In addition to issues associated with the effects of MUs on damage identification accuracy, one of the drawbacks of most conventional SHM ML techniques is the inability to use transfer knowledge from one structure under specific damage scenarios to another, similar, structure in a population, potentially one with different damage scenarios. Difficulty associated with transferring a developed and trained algorithm within a population of structures has motivated the SHM community to incorporate Population-Based SHM (PBSHM). Recently developed TL methods and their applications to SHM can provide systematic approaches to knowledge transfer within a population of structures [3,23]. Traditionally, the main assumption in implementing these methods is that training and testing data are selected from the same distribution [24]. To overcome this limitation, TL methods generalize the classifier trained for one structure to be applicable to another structure. In the context of SHM, TL methods have been used for image processing, computer vision, and pattern recognition. Gardner et al. [3] proposed a PBSHM by employing TL methods in the form of Domain Adaptation (DA). They assessed TL performance with respect to labeling a target domain using Transfer Component analysis (TCA) [25], Joint Domain Adaptation (JDA) [26], and Adaptation Regularization-based TL (ARTL) [27] in classification-type problems for homogeneous- and heterogenous-type populations. The efficiency of each TL method was examined for two heterogenous populations. The first case population encompassed two numerical simulations of a three degree of freedom structure, each with different geometric and material properties, with one simulation being the source and the other the target domain. The second case involved using a numerical simulation of a structure as the source domain and an experimental replica as the target domain. Recently, Zhang et al. [28] proposed a TL-based method and Bayesian model updating (BMU) to reduce the effect of MUs on model updating performance. Modal parameters, including normalized frequency change ratios and mode shapes, were used as features. They utilized ARTL to transfer knowledge from a source domain consisting of an eight-floor numerically simulated structure to a target six-floor experimental structure. Zhang et al. [29] used JDA as a TL method to map wave signals from one plate to another and a convolutional long short-term memory (ConvLSTM) network to learn mapping relationships from the source plate so that the damage image was detected in the targeted plate. Yan et al. [30] developed a structural anomaly detection framework using the transmissibility function and statistical threshold selection, and examined its robustness against uncertainty. Mei et al. [31] demonstrated the better performance of the Bhattacharyya distance-driven algorithm for novelty detection against transmissibility functions that follow Gaussioan distribution.

This study was motivated because research shows that a trained source domain TL-based classifier can be generalized to detect deficiencies in unlabeled target domains, and, when a JDA–kernel method is implemented, higher accuracy target domain label predictions are obtained [3]. To date, research regarding TL-based SHM, while promising,

has focused on experimental and numerical simulations under controlled environments. Therefore, a need to expand this approach to bridges and examine its effectiveness for actual structures under actual load with larger feature space exists. In addition, coupling this TL method with POD provides a robust damage identification framework with minimum sensitivity to noises.

This study focused on developing and studying a TL approach for transferring knowledge from a base, a modeled existing railway bridge class with known labels in the feature space, to a modeled bridge class with realistic MU and unknown labels in the feature space. Bridge FE models subjected to real train loadings measured during passages over the actual bridge under simulated damage scenarios were used to validate the TL approach for bridge damage identification. JDA coupled with a linear kernel, herein referred to as JDA–kernel, was implemented to map between POMs of the two bridge models. The derived relationship and the Kernel Nearest Neighbor (KNN) approach were then implemented to obtain POM labels for a target bridge with unknown labels. The resulting JDA–kernel approach for damage detection and intensity identification was evaluated using three scenarios: known damage intensity (DI) and unknown damage location (DL); known DL and unknown DI; and unknown DL and DI.

## 2. TL-Based Damage Identification

Previous research has shown that a TL approach can generalize trained damage detection knowledge from one structure to a group of similar structures [23,32]. To use ML methods to train a model for the damage classification of a structure, training and testing data distributions are assumed to be the same. This assumption might be violated for a population of structures with different features. This is the main motivation behind using a TL-based approach. To transfer the knowledge from one structure to another using this approach, the data distributions are mapped from a source domain to a targeted domain, thereby extensively reducing the computational effort needed for retraining. It is understood that the type of transferred knowledge and source to target structure mapping method relies on the level of similarities between the structures in the two mapped domains [32]. This research focused on a set of simulated experiments from a modeled bridge that are nominally identical. Therefore, the mapping used for this TL problem incorporates consistency in both the feature and label spaces.

### 2.1. SVD for Damage Detection

POD is a statistical method adopted for a wide variety of applications, including damage identification in structural systems. Orthogonal bases are ordered based on energy content [22] and, in the context of structural analysis, response data are stored in the form of a snapshot matrix. To obtain corresponding POMs, the SVD of the snapshot matrix is calculated as:

$$\boldsymbol{U} = \boldsymbol{L}\,\boldsymbol{\Sigma}\boldsymbol{R}^{T} \tag{1}$$

where $\boldsymbol{U}$ presents the snapshot matrix; $\boldsymbol{L}$ and $\boldsymbol{R}$ denote matrices containing left and right singular vectors, respectively; and $\boldsymbol{\Sigma}$ is a diagonal matrix containing singular values. Damage is identified using the left singular vectors, $\boldsymbol{L}$, of the snapshot matrix, which are the POMs [14,33]. The first POM is commonly considered for damage feature extraction.

### 2.2. TL Using JDA–kernel

To investigate TL efficiency in transforming knowledge from the source bridge to the target bridge, the JDA–kernel method was implemented. This effective TL method was developed for image processing and pattern recognition applications [26] and it was selected for this study because of its demonstrated robustness in transforming knowledge between domains with different marginal and conditional distributions [26] and its classification performance effectiveness [23]. The main goal of JDA is to minimize distances between the joint and conditional distributions for the source and target to find the optimum mapping function [26]. The linear kernel is incorporated into the JDA algorithm to transfer feature

components for the entire source domain into a kernel space for mapping purposes. The KNN classifier is then used to predict labels for part of the features in the target domain reserved for testing.

Two important aspects of the JDA–kernel are its domain and task [24]. A domain is denoted by $\mathcal{D} = \{\mathcal{X}, P(X)\}$ and contains a feature space $\mathcal{X}$ and a marginal probability distribution $P(X)$ of a feature $X$, where $X$ represents the data matrix and is a sample set from feature space $\mathcal{X}$. $X$ is defined as $X = \{x_i\}_{i=1}^{n} \in R^{m \times n}$, where $n$ denotes the total number of samples in the domain and $m$ represents the number of observations. A task is represented by $\mathcal{T} = \{\mathcal{Y}, f(\cdot)\}$ and consists of the label space $\mathcal{Y}$ and the classifier $f(\cdot)$. The classifier $f(\cdot)$ is also defined as the conditional distribution function denoted by $P(y|X)$, where $y \in \mathcal{Y}$. Consider a source domain $(D_s)$ and task $(T_s)$ and a target domain $(D_t)$ and task $(T_t)$. For homogeneous TL, it is assumed that the features and the label spaces between the domains are the same: $\mathcal{X}_s = \mathcal{X}_t$ and $\mathcal{Y}_s = \mathcal{Y}_t$.

DA is a subset of transductive TL and is used to improve the target predictive function utilizing knowledge obtained from $D_s$ and $T_s$ [3]. DA also assumes $\mathcal{X}_s = \mathcal{X}_t$ and $\mathcal{Y}_s = \mathcal{Y}_t$ while $P(X_s) \neq P(X_t)$ and $P(y_s|X_s) \neq P(y_t|X_t)$. For a case with single source and target domains, assuming that no label is available in the target domain, the domains are defined as [26]:

$$\mathcal{D}_s = \{(x_1, y_1), \ldots, (x_{n_s}, y_{n_s})\} \tag{2}$$

$$\mathcal{D}_t = \{x_{n_s+1}, \ldots, x_{n_s+n_t}\} \tag{3}$$

where: $n = n_s + n_t$, and $n_s$ and $n_t$ are the size of the samples available from the source and target domains, respectively.

For nonlinear mapping, the problem is kernelized using a Reproducing Kernel Hilbert Space (RKHS) and source and target joint and conditional distributions are computed in a new space. The kernel is defined to map $X$ to a corresponding $\phi(X)$ in the kernel space: $K(x_i, x_j) = \langle \phi(x_i), \phi(x_j) \rangle$, where $K(X, X) \in R^{(n_s+n_t) \times (n_s+n_t)}$. A k-dimensional orthogonal transformation matrix, $A$, in the kernel space for features $X$ is determined using Kernel Principal Component Analysis (PCA). Using PCA, the variance of matrix $A^T X$ is maximized by solving the following eigenvalue problem:

$$KHK^T A = A\Phi, \tag{4}$$

where $H$ denotes the centering matrix $H = I - \frac{1}{n} \mathbf{1}$, calculated using $I$ the Identity matrix, and $\mathbf{1}$ is an $n \times n$ matrix of ones; $\Phi$ represents the $k$ largest eigenvalues; and $A$ contains corresponding eigenvectors of the decomposed matrix, $KHK^T$. Upon determining $A$, the optimal transformation matrix, $Z$, is calculated as [26]:

$$Z = A^T K \tag{5}$$

Maximum Mean Discrepancy (MMD) is a kernel-based approach used to compute differences between marginal ($P(X_s)$ and $P(X_t)$) and conditional distributions ($P(y_s|X_s)$ and $P(y_t|X_t)$) in the source and target domains. This distance is calculated using the summation of the distances between sample means of source and target domain [26]. Distances for marginal distributions in the feature space with kernelized data are calculated as:

$$\left\| \frac{1}{n_s} \sum_{i=1}^{n_s} Z_i - \frac{1}{n_t} \sum_{j=n_s+1}^{n_s+n_t} Z_j \right\|^2 = tr\left(ZM_0Z^T\right) \tag{6}$$

where $M_0$ is the MMD matrix calculated as:

$$(M_0)_{ij} = \begin{cases} \frac{1}{n_s n_s}, & \text{for } x_i, x_j \in \mathcal{D}_s, \\ \frac{1}{n_t n_t}, & \text{for } x_i, x_j \in \mathcal{D}_t, \text{ or} \\ \frac{1}{n_s n_t} \end{cases} \tag{7}$$

Since the target domain is not labeled, JDA uses pseudo labels ($y_t{}^*$). The modified distance considering both marginal and conditional distributions is calculated as:

$$\left\| \frac{1}{n_s} \sum_{i=1}^{n_s} \mathbf{Z}_i - \frac{1}{n_t} \sum_{j=n_s+1}^{n_s+n_t} \mathbf{Z}_j \right\|^2 = tr\left( \mathbf{Z} \mathbf{M}_c \mathbf{Z}^T \right) \tag{8}$$

where $\mathbf{M}_c$ is the MMD matrix including class labels and it is calculated as:

$$(M_c)_{ij} = \begin{cases} \frac{1}{n_s^{(c)} n_s^{(c)}}, & \text{for } \mathbf{x}_i, \, \mathbf{x}_j \, \in \, D_s{}^{(c)} \\ \frac{1}{n_t^{(c)} n_t^{(c)}}, & \text{for } \mathbf{x}_i, \, \mathbf{x}_j \, \in \, D_t{}^{(c)} \\ \frac{1}{n_s^{(c)} n_t^{(c)}}, & \text{for } \begin{cases} \mathbf{x}_i \in \, D_s{}^{(c)}, \, \mathbf{x}_j \, \in \, D_s{}^{(c)} \\ \mathbf{x}_j \in \, D_s{}^{(c)}, \, \mathbf{x}_i \, \in \, D_s{}^{(c)} \end{cases}, \text{ or} \\ 0. \end{cases} \tag{9}$$

To constrain the problem, the regularization parameter, $\lambda$, is defined and the optimization problem is formed as follows:

$$\min_{A^T K H K^T A = I} \sum_{c=0}^{C} tr\left( A^T K M_c K^T A \right) + \lambda \|A\|^2 \tag{10}$$

By defining the Lagrangian function, this optimization problem is solved as an eigenvalue problem that contains the $k$ smallest eigenvalues:

$$\left( K \sum_{c=0}^{C} M_c K^T + \lambda I \right) A = K H K^T A \Phi \tag{11}$$

The $k$-dimensional optimal adaptation matrix, $A$, is a transformed matrix in the feature space and is further used for classification purposes. The JDA–kernel algorithm and KNN classifier are repeated several times until convergence happens. After each iteration, the performance of the algorithm is measured by comparing the true target labels ($y_t$) with the obtained pseudo label ($y_t{}^*$). Once the algorithm is converged, the optimized transformed matrix and corresponding predicted labels for target domain ($\hat{y}_t$) are obtained.

## 3. Validating TL for Bridge Damage Detection Using Simulated Experiments

### 3.1. Railway Bridge Study

The current study initially investigated a multi-span, in-service, steel railway bridge in central Nebraska, with a simply supported truss span being the focus. The bridge carries two tracks and its geometric and material properties are explained in more detail elsewhere [14,22]. The studied truss span is 44.7 m long and includes six panels. The floor system consists of timber ties resting on stringers that are connected to floor beams. The floor beams, in turn, distribute loads to truss lower chord panel points. Stringers are spaced 2.15 m center-to-center with tracks spaced 3.95 m center-to-center. Isometric, plan, and elevation views of the examined truss span, which also details instrument locations used for field testing, are shown in Figure 1.

To perform damage identification and extract damage features, strain time histories from bridge response to train passages were obtained from FE models. Stringer-to-floor beam connections were selected as the locations of interest because these spots were reported to be one of the most critical locations in terms of fatigue-induced damages in steel riveted railway bridges [22]. The locations of interests are depicted in Figure 1b and consisted of 20 locations that corresponded to strain transducer placement on the actual structure [34].

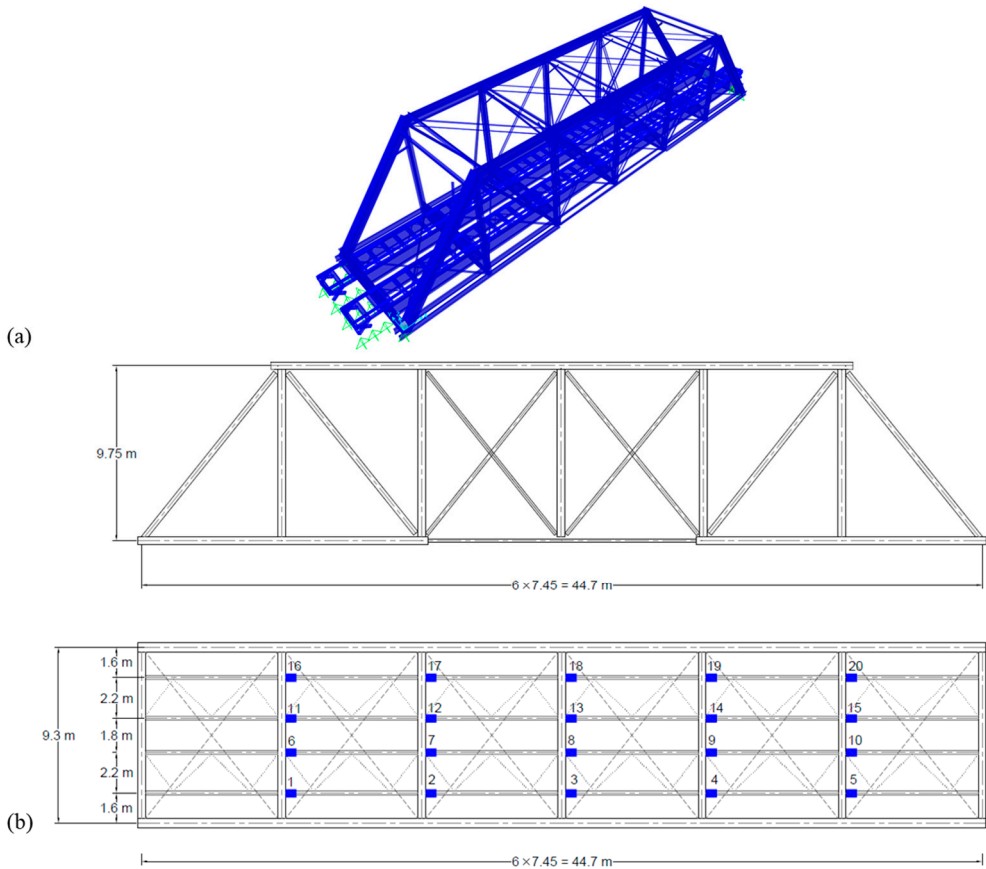

**Figure 1.** (**a**) Isometric view, truss span; (**b**) plan and elevation views and strain transducer locations.

### 3.2. FE Modeling

Numerical models of the bridge were developed using SAP2000 [35]. The Open Application Programming Interface (OAPI) was used to assist with algorithm development that integrates MATLAB [36] with SAP2000 to automatically initialize numerical simulations. Model performance was validated against field data. To investigate potential damage at locations of interest identified in Figure 1, stringer connections were modeled using rotational springs, and connection damage levels, termed DI, were simulated by reducing their rotational stiffness [22].

Structural response to the recorded train loads in a simulated healthy bridge was obtained in the form of strain time histories and validated against the measured data. For each train that traversed the bridge, the strain time histories at each sensor location were stored in the snapshot matrix. The Root Mean Square (RMS) means of the snapshot matrix was used to sort the train loads. The train events were then divided into four groups based on their RMS and the last group with a higher RMS (heavier trains) was involved in this study [14,22]. Validated numerical models were used to examine structural response to several modeled damage scenarios representing fatigue cracks at stringer-to-floor beam connections [37]. Stringers connections were modeled using rotational springs and deficiency due to fatigue cracks was simulated by a gradual decrease in rotational stiffness at the connections. Rotational stiffness reductions were in 10% increments, starting with 0% (uncracked) to 100% (completely cracked). POMs were developed for multiple field recorded loading events for each damage scenario. At total of 20 DL and 24 train load configurations were considered, resulting in 4800 damaged and 24 healthy scenarios being modeled.

### 3.3. Simulated Experiments

A previous study [22] showed that, for local damage, POMs in the healthy structure, $\varphi^h$, are related to POMs in the damaged structure, $\varphi^d$, as:

$$\Delta\varphi = \varphi^d - \varphi^h \tag{12}$$

where $\Delta\varphi$ is defined as a feature in the damage detection framework for any model with existing MU. They showed that $\Delta\varphi$ for a target domain can be approximated using $\Delta\varphi$ from a source domain and the estimated POMs using this method can be used for training the ANN. The neural network architecture consists of an input layer, a hidden layer with sigmoid activation function, and an output layer with identity matrix as an activation function. For more details regarding the ANN training algorithm, the readers are referred to [14].

In this study, the performance of JDA–kernel for TL was investigated for a set of six simulated cases from a modeled bridge, as illustrated in Table 1 [22]. Simulated experiments contain a set of modeled structures with identical geometry, materials, and topology and, therefore, have consistent feature and label spaces [23]. Stringer-to-floor beam connection end fixity ratio significantly affects the stress time histories results. In addition, due to the high variation of end fixity ratios, the difference between the actual and modeled ratios were high [37,38]. Therefore, the end fixity ratio was selected as MU. Specific stringer-to-floor beam end fixity ratios were assigned to each model. Models in Table 1 had different levels of uncertainties represented by assigned spring coefficients for stringers and are denoted with a percentage from the base model spring coefficient. The uncalibrated base model from these experiments with no MU was identified as $M$. Its calibrated version was denoted as $M_0$, with other models having various MU levels compared to these base models. It should be noted that the same MU was assigned to both $M_0$ and $M_1$. The discrepancies between these two models is attributed to features generated for training the damage identification framework. $M_0$ was trained using $\varphi$ obtained from simulation, whereas $M_1$ used estimated $\varphi$ with $\Delta\varphi$ for the same purpose.

**Table 1.** Simulated experiments.

| Model | Assignment | Normalized Rotational Spring Coefficient |
|:---:|:---:|:---:|
| $M_0$ | Uniform increase +80% in $M$ | 1.80 |
| $M_1$ | Uniform increase +80% in $M$, estimated $\varphi$ with $\Delta\varphi$ | 1.80 |
| $M_2$ | Uniform decrease $-50\%$ in $M$ | 0.5 |
| $M_3$ | Random, $\pm50\%$ in $M$ | Between 0.53 and 1.45 |
| $M_4$ | Random, $\pm25\%$ in $M$ | Between 0.76 and 1.25 |
| $M_5$ | Random, $\pm100\%$ in $M$ | Between 0.23 and 1.92 |

### 3.4. POMs

Strains from each model scenario in Table 1 were stored in a snapshot matrix and used to extract POMs. Healthy POMs were calculated from the snapshot matrix for each simulation using SVD, and the first POM was then extracted. The first POMs for $M_0$ were trained for damage identification using ANN [14]. In this study, simulated experiments $M_1$ to $M_5$ from Table 1 were trained for damage scenarios using $\Delta\varphi$ as feature, and the estimated first POMs were calculated at each location shown in Figure 1 using the developed framework from Rageh et al. [22]. Figure 2 plots the average of the first POMs for the different models, each without imposed damage. These plots indicate that the presence of MU at different levels in the FE model affect the first POM variations and necessitate the use of a TL approach for transfer knowledge across different simulated experiments. These POMs were used as features in the TL algorithm, as discussed in the following section, using the JDA–kernel method. The JDA–kernel's ability to map trained knowledge from $M_0$ to other targeted models was then evaluated.

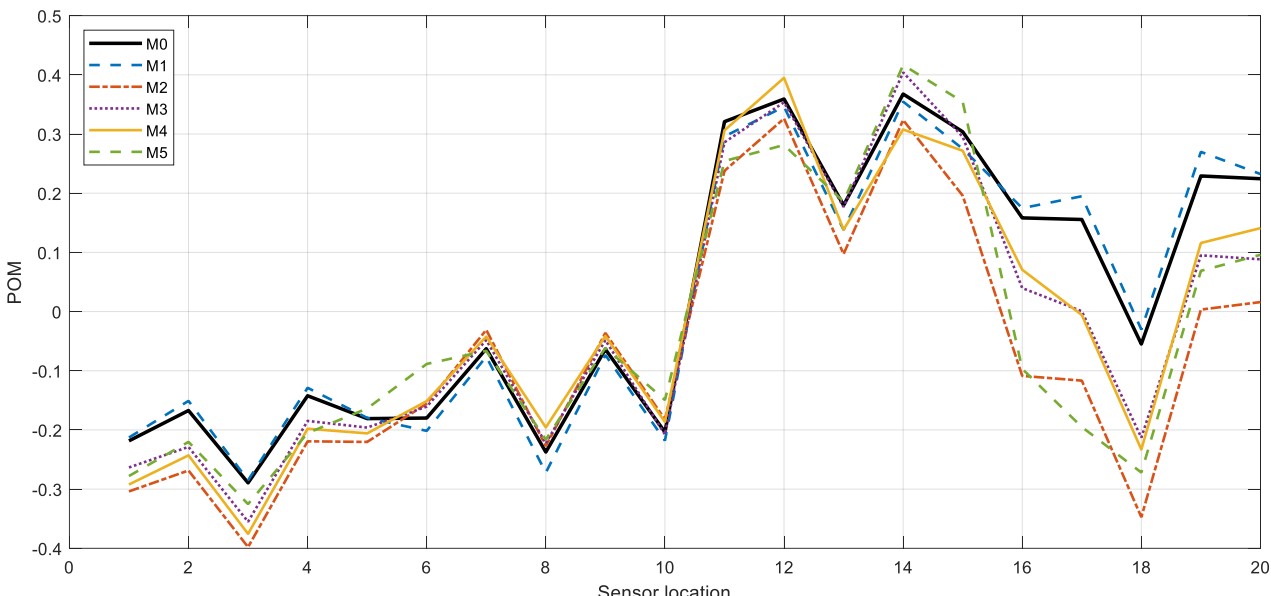

**Figure 2.** Healthy average first POMs for each model.

## 4. TL Using Coupled POD and JDA–kernel

Before implementing the JDA–kernel method, two models were selected from the experiment set to map knowledge from one model, identified as the source domain, to another, target domain, model to predict labels for the target domain. The base model, $M_0$, was selected as the source domain, and each model with MU from Table 1 as the target domain. Each model had eleven damage labels, one for the healthy state (0% damage) and ten imposed damage levels with different intensities $DI = \{10, 20, \ldots, 100\%\}$ at different locations $DL = \{1, 2, \ldots, 20\}$, identified in Figure 1b. Feature spaces were the first POMs. The base model ($M_0$) was simulated for all 4824 scenarios using SAP2000 OAPI and validated using field measured data from the bridge.

The JDA–kernel method and KNN classifier were implemented for damage identification in the target domain using trained knowledge from the source domain. Each TL implementation assumed that estimated POMs for source and target domains from [22] were classified and labels for the source domain known. Predicted target domain labels from this supervised TL implementation were compared with those from previously known labels to determine their accuracy. Since the JDA–kernel generates pseudo labels, the algorithm was implemented over several iterations until it converged.

Three scenarios were investigated. Scenario 1 was where DI was known, and the algorithm determined DLs. The feature space in both the source and target domains contain matrices of the first POMs where each column represented the associated first POM for a particular DL, where $x_{n_s} \in R^{m \times n_s}$ and $n_s = 480$, and the label space included $\mathcal{Y}_s = \mathcal{Y}_t \in \{1, 2, \ldots, 20\}$. Scenario 2 identified DIs in the target domain with DL being known. Here, the feature space in both the source and target domains contain matrices of the first POMs where each column represented the associated first POM for a particular DI, where $x_{n_s} \in R^{m \times n_s}$ and $n_s = 264$, and the label space included $\mathcal{Y}_s = \mathcal{Y}_t \in \{0, 1, 2, \ldots, 10\}$. For Scenario 3, both DIs and DLs were unknown and the algorithm was applied to the entire dataset in the source and target domains to determine target domain labels. Categorization was performed for various DIs and the label space was defined as $\mathcal{Y}_s = \mathcal{Y}_t \in \{0, 1, 2, \ldots, 10\}$. The feature space was defined as $x_{n_s}$, where $x_{n_s} \in R^{m \times n_s}$ and $n_s = 4824$ (24 features for healthy case and 480 features for each DI). In this scenario, redundant healthy POMs were added to the feature space so that the same number of features from each damage state were included to prevent the training process from underestimating healthy states. This provided $n_s = 5280$.

Cross-validation of the JDA–kernel algorithm was implemented using the k-fold method [39]. Four partitions were used, with one partition being reserved for cross-validation and training being carried out using the remaining three partitions. Hyperparameter tuning was also implemented using the grid search method [40] to find optimum values for the number of transferred components ($k$) and the regularization parameter ($\lambda$) used in each simulation. Figure 3 details the TL process. As illustrated in the figure, the process consisted of four steps. Preparing features and defining source and target models were addressed in Step 1. JDA–kernel implementation, cross-validation, classifier training, and hyperparameter tuning were implemented in Step 2. In Steps 3 and 4, the JDA–kernel and classifier were reimplemented using partial data and entire data sets, respectively, from optimized parameters obtained in Step 2. The score in Step 3 and 4 represents the calculated mean accuracy of the JDA–kernel and classifier by applying the optimized parameters in the algorithm.

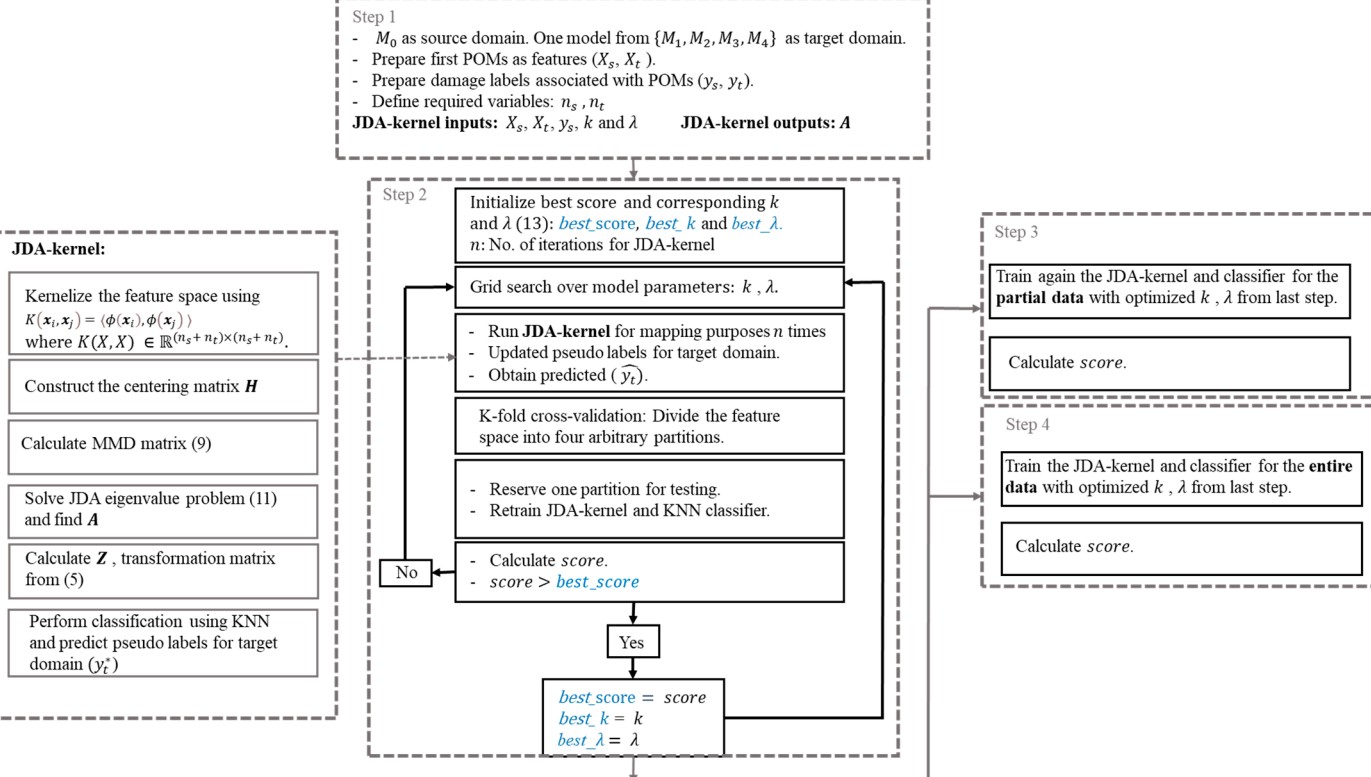

**Figure 3.** TL process for training, cross validation, and hyperparameter tuning.

## 5. Results and Discussion

In this section, results from the three previously outlined TL scenarios are summarized and the effectiveness with which the TL approach identifies damage location and intensity assessed. The features from the source domain, model $M_0$, were mapped to various target domains using the TL algorithm, with each target being one of the models $\{M_1, M_2, M_3, M_5\}$ representing a different MU. TL results for $M_4$ are not shown due to POM similarities between $M_3$ and $M_4$. All comparisons were for cases where applied loads were assumed unknown.

### 5.1. Scenario 1: DI Known, DL Unknown

Figure 4 shows confusion matrices used to compare predicted labels from transferred knowledge using the JDA–kernel to "true" values, which represent the DLs (see Figure 1b), for target domains $M_1, M_2, M_3, M_5$ at $DI = 80\%$. Confusion matrices are used to compare predicted labels obtained using the TL algorithm to true labels for the damage states with

axes labels for the matrices corresponding to damage states labels from $\mathcal{Y}_t \in \{1, 2, \dots, 20\}$. It is observed that DL labels obtained using the JDA–kernel for the target domain at most locations agree very well with the assigned labels for all models. Prediction error increased in DLs 2, 4, 7, and 9. These errors were attributed to average POMs being of similar magnitudes for the examined DLs and, as a result, the classifier could not effectively distinguish between them. Coupled POD JDA–kernel DL detection performance for each model is summarized in Table 2, which shows the accuracy of the algorithm for all models and for a DI of 80%. Overall, the algorithm was shown to perform well for identifying DLs for this DI.

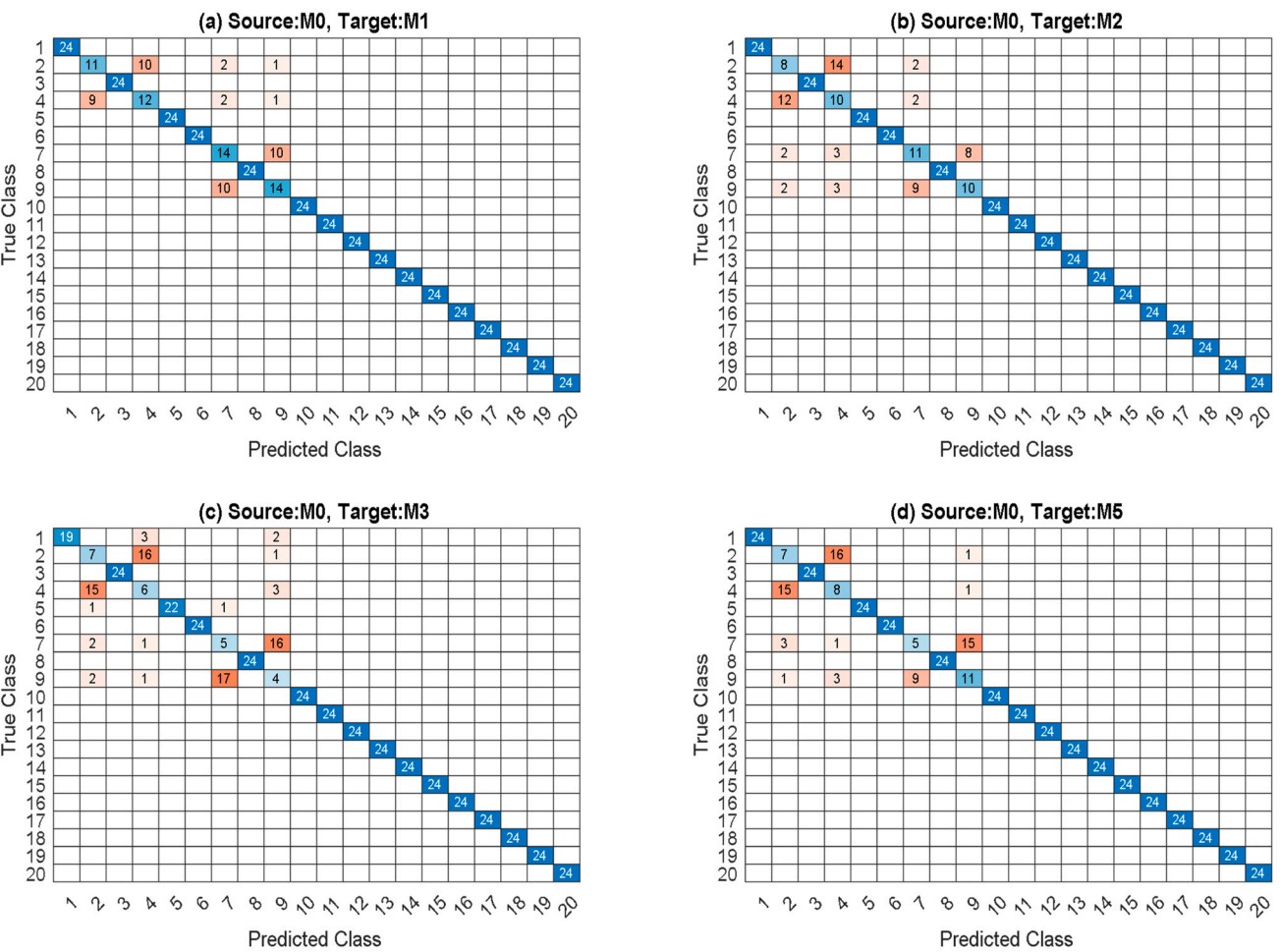

**Figure 4.** Confusion matrix, $M_0$ for DI = 80%: (**a**) $M_1$, (**b**) $M_2$, (**c**) $M_3$, and (**d**) $M_5$.

**Table 2.** Coupled POD JDA–kernel Accuracy, DI = 80%.

| Target Model | Accuracy, Cross-Validation (%) | Accuracy, Entire Data (%) |
|:---:|:---:|:---:|
| $M_1$ | 90 | 100 |
| $M_2$ | 88 | 89 |
| $M_3$ | 87 | 85 |
| $M_5$ | 87 | 94 |

Prediction results for DIs of 40, 60, 80, and 100% are shown in Figure 5. $M_1$ was arbitrarily selected as the target model for all cases as M1 results were representative of other models. As expected, JDA–kernel accuracy increases in higher DIs, with similar DL prediction patterns being observed, as for the confusion matrices shown in Figure 5. Table 3

summarizes JDA–kernel accuracy results in $M_1$ at different DIs. The table details improved accuracy for higher levels of DI. Similar patterns were observed for the other models.

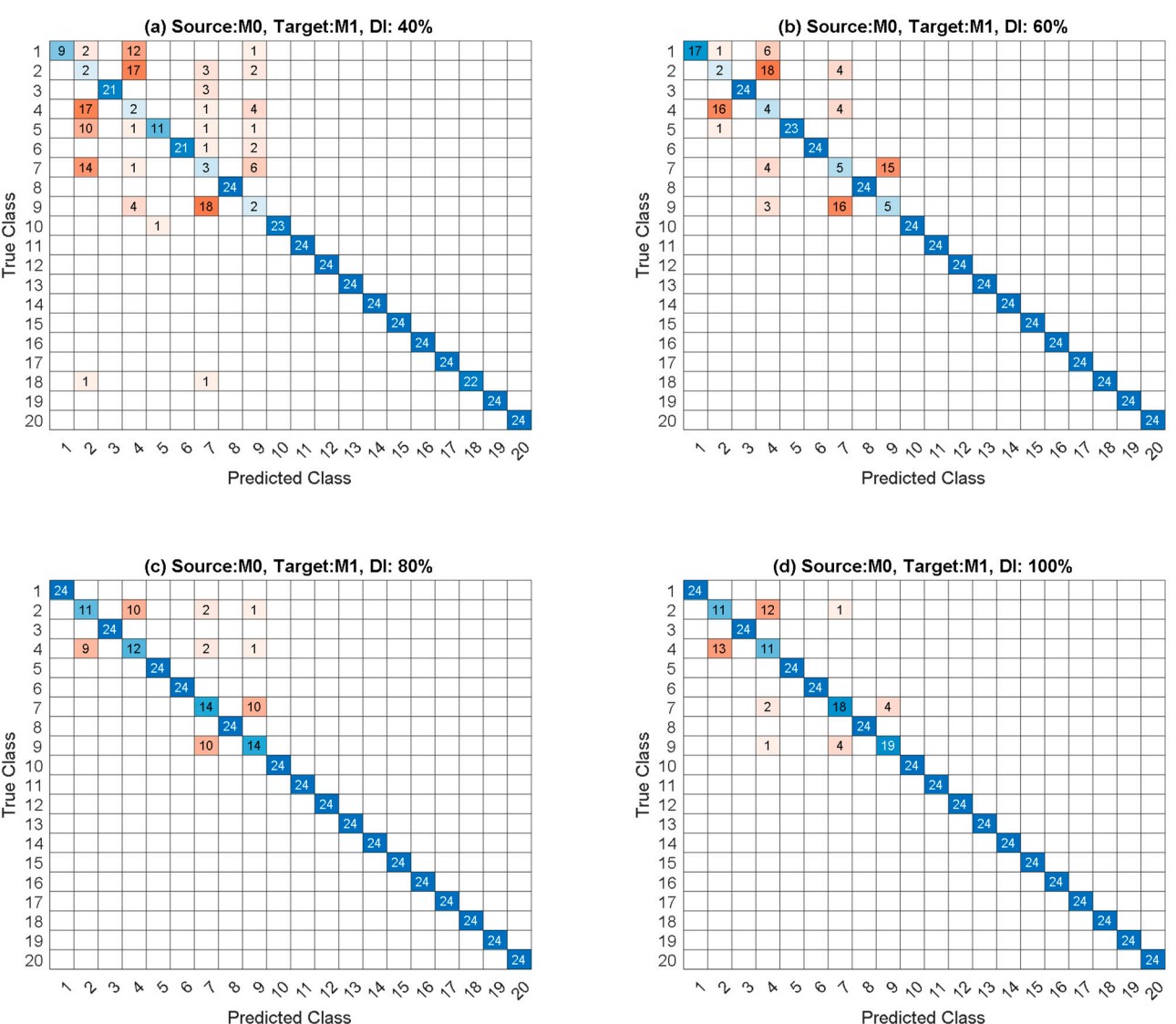

**Figure 5.** Confusion matrix for model $M_1$ as target at DIs (**a**) 40%, (**b**) 60%, (**c**) 80%, and (**d**) 100%.

**Table 3.** Coupled POD JDA–kernel Accuracy, Target: $M_1$.

| DI (%) | Accuracy, Test Data (%) | Accuracy, Entire Data (%) |
|---|---|---|
| 40 | 72 | 92 |
| 60 | 85 | 98 |
| 80 | 90 | 100 |
| 100 | 94 | 99 |

### 5.2. Scenario 2: DL Known, DI Unknown

The JDA–kernel was further examined via identification of DI labels for target domain ( $\mathcal{Y}_t \in \{0, 1, 2, \ldots, 10\}$ ) at select locations when DL was assumed known. Figure 6 demonstrates the algorithm performance for target models $M_1$, $M_2$, $M_3$, $M_5$ at representative DL 18 (see Figure 1b). As illustrated in the figure, JDA–kernel performance increases with increasing damage intensity in each model. In addition, false positives occur at intensities with magnitudes close to expected intensities for all presented target models, indicating promising performance of the JDA–kernel method. The JDA–kernel DI estimation

performance at other DLs was investigated and similar patterns were observed. Table 4 presents JDA–kernel accuracies associated with results observed in Figure 6. As shown, the JDA–kernel algorithm could adequately identify DI at DL 18 for the studied models.

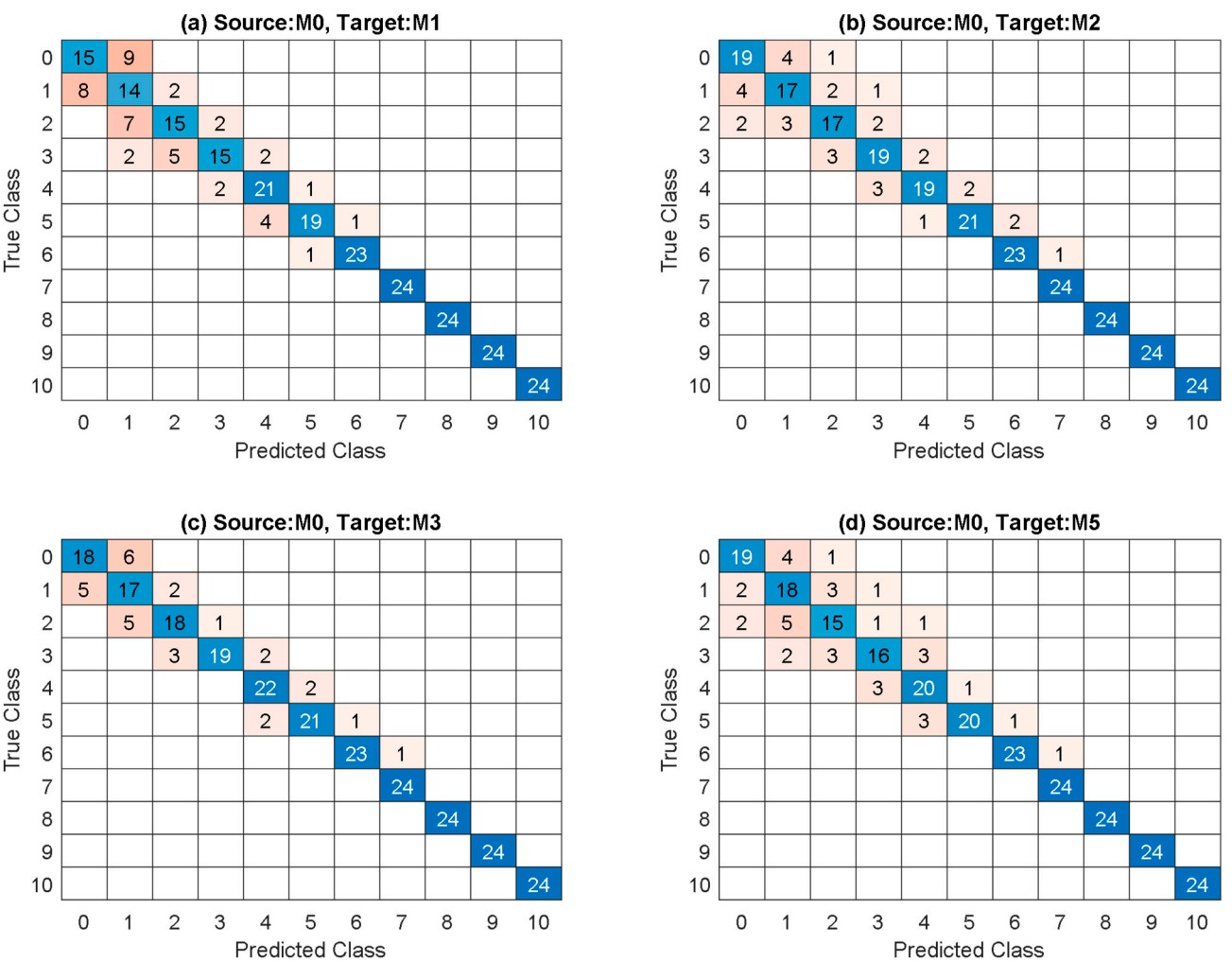

**Figure 6.** Confusion matrix at DL 18 (see Figure 1b): (**a**) $M_1$, (**b**) $M_2$, (**c**) $M_3$, and (**d**) $M_5$.

**Table 4.** Accuracy results for coupled POD JDA–kernel, DL 18.

| Target Model | Accuracy, Test Data (%) | Accuracy, Entire Data (%) |
| :---: | :---: | :---: |
| $M_1$ | 85 | 89 |
| $M_2$ | 92 | 84 |
| $M_3$ | 90 | 93 |
| $M_5$ | 90 | 87 |

$M_1$ was then assigned as the target domain and influence of the known DL on the JDA–kernel performance investigated, with results shown in Figure 7. Four representative DLs were selected, with DLs 13 and 18 located beneath the loaded track and DLs 3 and 8 not beneath the loaded track. It is observed that, for the representative DLs shown in the figure, accuracy again improves at higher Dis. Similar results patterns were observed in other locations. Table 5 summarizes these results. Improved accuracy was observed in DL 13 and 18 when compared against DL 3 and 8. POM variations were more significant in DLs 13 and 18 due to their locations relative to applied loads.

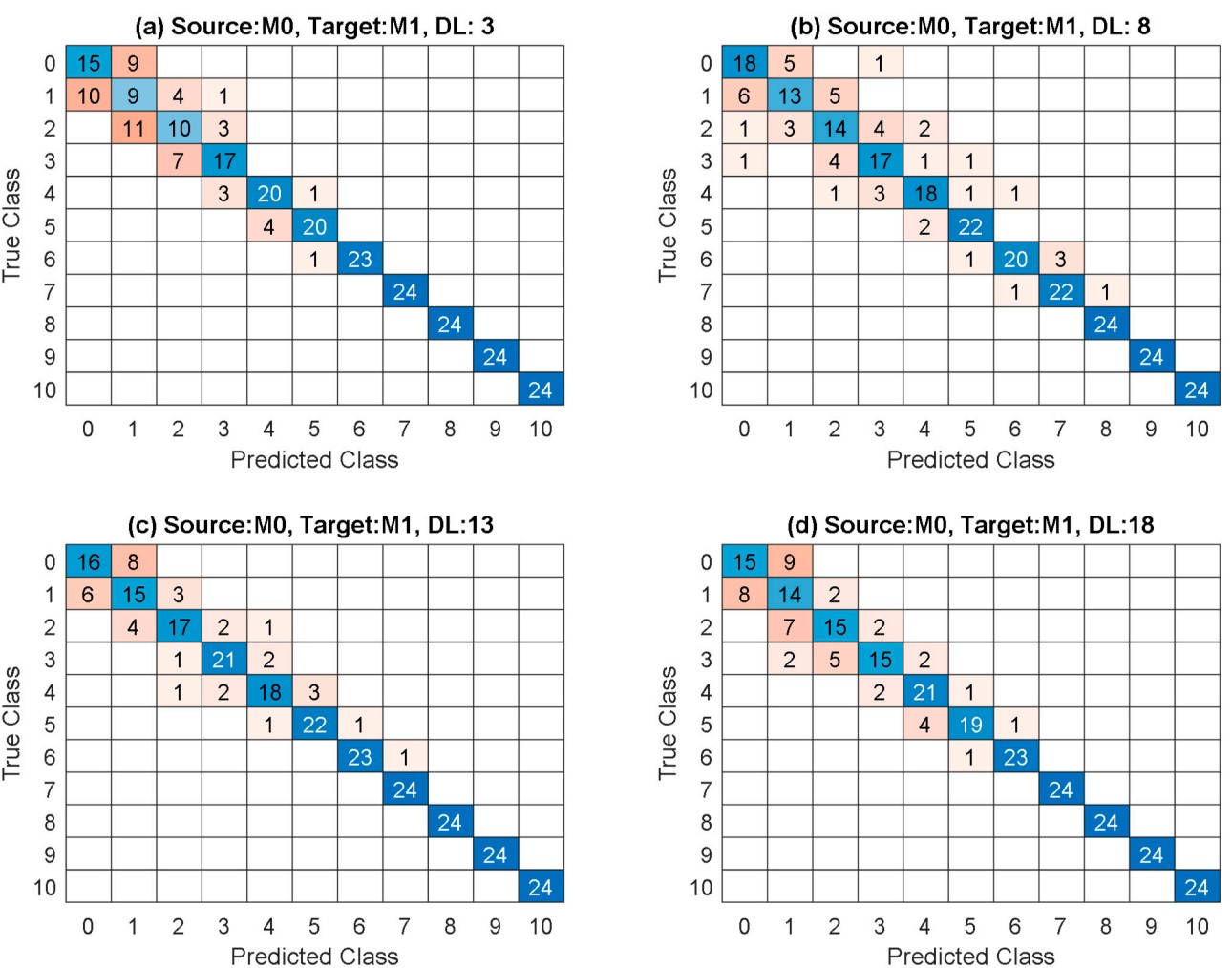

**Figure 7.** Confusion matrix for model $M_1$ as target at DLs (**a**) 3, (**b**) 8, (**c**) 13, and (**d**) 18 (see Figure 1b).

**Table 5.** Accuracy results for coupled POD JDA–kernel, Target: $M_1$.

| DL | Accuracy, Test Data (%) | Accuracy, Entire Data (%) |
|---|---|---|
| 3 | 82 | 81 |
| 8 | 81 | 88 |
| 13 | 88 | 93 |
| 18 | 85 | 89 |

### 5.3. Scenario 3: DL and DI Unknown

The effectiveness of JDA–kernel was examined using the entire dataset by simultaneously varying loads, DL, and intensity. Categorization was performed for various DIs, and results are shown in Figure 8 for all models. When compared to the previous two scenarios, prediction error associated with each model increased due to an increase in the number of unknown variables. It was observed that JDA–kernel performance with respect to damage label prediction improved in higher DIs. It was of interest to compare using JDA–kernel for TL against using KNN without TL for each target model, because this comparison demonstrates the successful effect of JDA–kernel caused by mapping the transfer components from source to target domain on TL performance. Results of the comparisons are shown in Table 6, and significant improvement in prediction accuracy was observed in the JDA–kernel TL method. The runtimes are also reported in Table 6. It is noted that JDA–kernel requires several iterations to improve the performance.

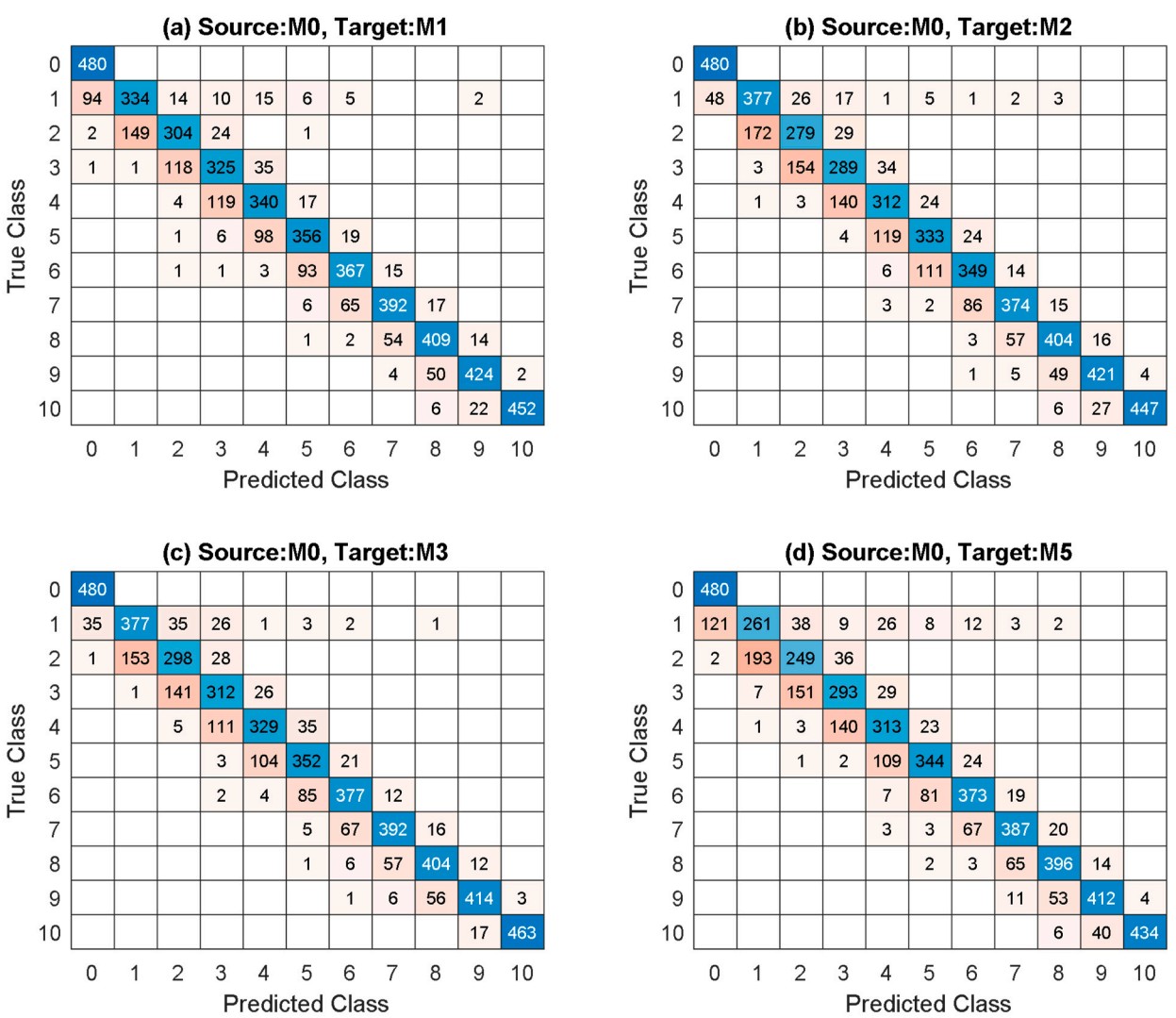

**Figure 8.** Confusion matrix associated with each model for unknown DLs and DIs and loading event. Source model $M_0$ and target model (**a**) $M_1$, (**b**) $M_2$, (**c**) $M_3$, and (**d**) $M_5$.

**Table 6.** Accuracy and runtime results comparison, unknown DLs, DIs, and loading.

| Target Model | Accuracy, KNN, without Applying TL (%) | Accuracy, JDA–kernel, Test Data (%) | Accuracy, JDA–kernel, Entire Data (%) | KNN without TL, Runtime (s) | JDA–kernel, Runtime (s) |
|---|---|---|---|---|---|
| $M_1$ | 35 | 79 | 86 | 0.3 | 1464 |
| $M_2$ | 12 | 77 | 76 | 0.3 | 1420 |
| $M_3$ | 18 | 80 | 75 | 0.3 | 1442 |
| $M_5$ | 12 | 75 | 73 | 0.3 | 1445 |

## 6. Conclusions

A TL-based POD damage identification method was developed and investigated as a potential damage detection framework that could potentially provide more accurate damage predictions in a target domain by transferring knowledge using data from a source domain. The method was investigated for simulated cases of bridges with consistent labels, which represented damage location and intensity, with the source domain being bridges with known labels and the target domain being bridges with unknown labels. This study examined the effectiveness with which TL estimated damage in a target domain based on data from a source domain.

The JDA–kernel TL method utilized a set of simulated experiments from a modeled bridge developed from a tested steel railway truss bridge, with experiments including different levels of uncertainty, to develop a TL-coupled POD framework for damage identification. A previously published POD-based approach for damage identification was used to extract POMs as damage features from each model [22]. The accuracy of the proposed approach was assessed using confusion matrices for predicted labels in each target model.

Findings indicated that:

- TL successfully identified DL labels for each target model. Label identification was less accurate at locations with similar POMs.
- TL was shown to be an effective method for identifying the DIs for the bridge and MUs that were studied. As expected, TL method accuracy improved at larger DIs.
- Live load position influenced TL method effectiveness, with more effective DI identification occurring in close proximity to loaded tracks.
- The JDA–kernel TL significantly improved damage identification when compared against KNN.

**Author Contributions:** Conceptualization, S.A. and S.E.A.; methodology, S.A. and S.E.A.; validation, S.A., S.E.A. and D.G.L.; formal analysis, S.A.; investigation, S.A.; data curation, S.A.; writing—original draft preparation, S.A.; writing—review and editing, S.A., S.E.A. and D.G.L.; visualization, S.A.; supervision, D.G.L.; funding acquisition, D.G.L. All authors have read and agreed to the published version of the manuscript.

**Funding:** This research was funded by the National Science Foundation, grant number 1762034, Spokes: MEDIUM: MIDWEST: Smart Big Data Pipeline for Aging Rural Bridge Transportation Infrastructure (SMARTI).

**Institutional Review Board Statement:** Not applicable.

**Informed Consent Statement:** Not applicable.

**Data Availability Statement:** Not applicable.

**Acknowledgments:** The authors gratefully acknowledge assistance, access, computing resources, data, and expertise provided by the University of Nebraska Lincoln's Holland Computing Center, Union Pacific, and Bridge Diagnostics Inc. in association with this project. The authors would also like to thank all anonymous reviewers for their constructive suggestions.

**Conflicts of Interest:** The authors declare no conflict of interest. The funders had no role in the design of the study; in the collection, analyses, or interpretation of data; in the writing of the manuscript; or in the decision to publish the results.

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
