# Peer review of "Bridge Health Monitoring Using Proper Orthogonal Decomposition and Transfer Learning"

_applsci, doi:10.3390/app13031935_

Round 1

Reviewer 1 Report

General comments:

The paper presents a damage identification method by coupling JDA-kernel-based transfer learning and POD, aiming at transferring knowledge obtained from a labeled source domain to a target domain without labels to reduce the computational effort needed for retraining. The performance of the proposed method is validated using a set of simulated experiments from a modeled bridge developed from a tested steel, railway, truss bridge, including different levels of uncertainty, results show that this method exhibits better accuracy for identifying both damage intensity and unknown damage location compared with the KNN-based method. Generally, the paper is well-written and develops a promising work in SHM.

Specific comments:

1.      Although the overall writing is good, there are still some confusing sentences that need to be clarified. For example, on Page 4 line 140, the author declared: “a limitation that is the main motivation behind using a TL-based approach because it maps data distributions from a source domain to a targeted domain, thereby extensively reducing computational effort needed for retraining.” However, I have no idea what you mean about the limitation.

Besides, in Fig. 4d and Fig. 6, the confusion matrices correspond to the damage identification results of scenario M5 instead of M4.

Please check the paper and clarify these confusing expressions and typos.

2.      In Table 1 six simulated cases with uncertainty are introduced, however, I have no idea how the uncertainty is introduced in experiments M1 and M2 as the normalized rotational spring coefficient is constant in these two cases. In addition, the author should consider if it is sufficient to simulate the uncertainty from modelling error, measurement noise, environmental variabilities, etc. in real SHM tasks by only the variation of the spring coefficient.

3.      In Page 5, it is assumed that no label is available in the target domain, however, the prediction of the proposed method is obtained by comparing the true target labels ?? with the obtained pseudo label ??. I am confused how to conduct this comparison as no label is available in the target domain.

4.      Similar to comment 3, in Section 3.4, the author said that the damage scenarios are trained using Δφ in simulated experiments M1 to M5. However, I wonder how to train a neural network using Δφ as there are no labels for these experiments which are said to be the target domain in the following section.

5.      In Section 2.2, the author said that JDA-kernel is used to minimize distances between the joint and conditional distributions for the source and target domains to find the optimum mapping function for knowledge transferring, and MMD is used as the distance in this paper. However, there are some studies in SHM using probabilistic distances to quantify the difference between two distributions, which have a better capability in accommodating the uncertainty from measurement noise and environmental variabilities than simply using the summation of the distances between sample means. The author is advised to include these works, which may help improve the performance of the proposed method (Structural novelty detection with Laplace asymptotic expansion of the Bhattacharyya distance of transmissibility and Bayesian resampling scheme; Structural anomaly detection based on probabilistic distance measures of transmissibility function and statistical threshold selection scheme).

Reviewer 2 Report

1) Some writing issues should be paid attention to. Typically, we do not include any references in the abstract.

2) transfer learning has already been used in the health monitoring of machine systems, such as [1]. Can some comments be made to highlight the novelty of this paper?

[1] Li, Chuan, et al. "A systematic review of deep transfer learning for machinery fault diagnosis." Neurocomputing 407 (2020): 121-135.

3) As far as the reviewer’s understanding, the novelty of this paper is the application of TL in the long-span structure in transportation engineering. The proposed method has the potential to be used in some other railway infrastructures like the track [2] and overhead lines [3]. It is recommended to indicate the potential value of this study in the introduction with some literature review.

[2] Malekjafarian, Abdollah, et al. "Railway track monitoring using train measurements: an experimental case study." Applied Sciences 9.22 (2019): 4859.

[3] Song, Yang, et al. "Contact wire irregularity stochastics and effect on high-speed railway pantograph–catenary interactions." IEEE Transactions on Instrumentation and Measurement 69.10 (2020): 8196-8206.

4) The format of texts in lines 132-134 is different from others. Please fix this issue.

5) Please give more detailed description of the train load acting on the bridge in the simulation model.

6) Another issue is regarding the validation of the simulation model. No evidence is provided to demonstrate the validity of the numerical simulation. Normally, the test of the proposed method should rely on a reliable simulation model or field test. Please discuss this issue.

7) How is the degradation reflected in your bridge model? Please give more details. It seems that the degradation modelling is not given sufficient details.

Round 2

Reviewer 2 Report

All my comments have been properly addressed. I think this paper can be published as a good contribution to the research field.

Author Response

Dear reviewer 2,

We would like to sincerely thank you for your time reviewing our manuscript. Your suggestions allowed us to greatly improve the quality of the manuscript. 

Sincerely,
Samira Ardani on behalf of all the authors

Reviewer 3 Report

The authors have addressed most of the reviewer's previous comments. However, the runtimes are still not reported, therefore the efficiency is not clearly demonstrated. 

Author Response

Dear reviewer 3,

Thank you for your comment on our manuscript. We agree with your comment, and we modified the manuscript to accommodate your comment. Please see page 18, line 432-433 and Table 6 of the revised manuscript.

Your suggestions allowed us to greatly improve the quality of the manuscript. We would like to sincerely thank you for your suggestions and constructive comments.

Sincerely,

Samira Ardani on behalf of all the authors
